# Smartphone Impostor Detection with Behavioral Data Privacy and Minimalist Hardware Support

## ABSTRACT

Impostors are attackers who take over a smartphone and gain access to the legitimate user's confidential and private information. This paper proposes a defense-in-depth mechanism that can detect impostors quickly with simple Deep Learning algorithms, which can achieve better detection accuracy than the best prior work which used Machine Learning algorithms requiring computation of multiple features. Different from previous work, we then consider protecting the privacy of a user's behavioral (sensor) data by not exposing it outside the smartphone. For this scenario, we propose a Recurrent Neural Network (RNN) based Deep Learning algorithm that uses only the legitimate user's sensor data to learn his/her normal behavior. We propose to use Prediction Error Distribution (PED) to enhance the detection accuracy. To make the on-device, real-time detection possible, we show how a minimalist hardware module, dubbed SID for Smartphone Imposter Detector, can be designed and integrated into smartphones for self-contained impostor detection. Experimental results show that SID can support real-time impostor detection, at a very low hardware cost and energy consumption, compared to other RNN accelerators.

## 1 INTRODUCTION

Smartphone theft is one of the biggest threats to smartphone users [28]. Impostors are defined as adversaries who take over a smartphone and perform actions allowed only for the legitimate smartphone owners. Impostor attacks breach the confidentiality, privacy and integrity of the sensitive personal information stored in the smartphone, and accessible online through the smartphone. As powerful attackers may already know, or can bypass, the legitimate smartphone user's password or personal identification number (PIN), can a defense-in-depth mechanism be provided to detect impostors quickly before further damage is done? Can this be done implicitly, using behavioral biometrics, like how a user moves or uses the phone?

The ubiquitous inclusion of motion sensors, e.g., the 3-axis accelerometer and the 3-axis gyroscope, in smartphones provides a great opportunity to capture a user's large and small motion patterns. In the literature, implicit smartphone user authentication with sensors is primarily modeled as a binary classification problem [5, 6, 11, 12, 18, 30, 31]. While these can also be leveraged as implicit impostor detection systems, the problem is these past work require both the legitimate user's sensor data and other users' sensor data for model training. This causes serious privacy issues as users must provide their sensitive behavioral data to a centralized training system. The privacy protection of a user's behavioral biometric data in implicit impostor detection (conversely, implicit user authentication) has not been investigated in the past, which we do in this paper.

The privacy of a user's biometric data can be preserved if it does not need to be sent to the cloud for training with other users'

data. Hence, we propose using only the legitimate user's data to train a Recurrent Neural Network (RNN) to learn the user's normal behavior. A large deviation of the currently observed behavior from the model's prediction indicates that the smartphone is not being used by the legitimate user, and hence it is probably being used by an impostor. To achieve high detection accuracy, we further propose comparing the model's prediction error distributions. We show that this can significantly improve the detection accuracy in one-class deep learning scenarios like ours.

To reduce the attack surface and the time taken for impostor detection, we design a small and energy-efficient hardware module for impostor detection on the smartphone. While previous ML/DL accelerators try to maximize the performance of one or a few specific ML or DL algorithms (e.g. RNN), which is only a part of the detection process, our goal is to support the end-to-end detection process and provide *sufficient performance at low cost and low power consumption*.

Our Smartphone Impostor Detector (SID) module is flexible in that it supports not only the best deep learning algorithms we found for impostor detection in both user scenarios (with and without other users' data for training) but also other Machine learning (ML) and Deep Learning (DL) algorithms. Furthermore, it accelerates the computation of empirical probability distributions and statistical tests. SID reuses functional units where possible to reduce hardware size and cost. It is also scalable for higher performance if more parallel datapath tracks are implemented. Programmability provides flexibility in selecting algorithms and adjusting trade-offs in security, privacy, usability and costs, e.g., execution time, memory requirements and energy consumption. Our key contributions are as follows:

- We propose privacy-preserving smartphone detection algorithms that detect imposters and implicitly authenticates users, while protecting their behavioral data privacy.
- We show that we can significantly improve the accuracy of a deep learning algorithm (LSTM or other RNN) for detecting abnormal users (i.e., impostors) by comparing prediction error distributions.
- When both the user and non-users' data are used for centralized (2-class) model training, we show that a simple deep learning algorithm, MLP, can outperform the previous best implicit user authentication algorithm, KRR with honed feature selection.
- We propose a new light-weight hardware module, SID, that is versatile, scalable and efficient for performing impostor detection without preprocessing or postprocessing on the CPU or other devices. This eliminates the storage of sensor data and the network exposure for data transmissions, thus significantly reducing user data exposure.

## 2 THREAT MODEL AND ASSUMPTIONS

Our threat model includes powerful attackers who can bypass the conventional explicit authentication mechanisms, e.g. password or personal identification number (PIN). For example, the PIN/password may not be strong enough. The attacker can actively figure out the weak pin/password by guessing or social engineering. Another example is the attacker taking the phone after the legitimate user has entered his password.

We assume the smartphone has common, built-in motion sensors, e.g. the accelerometer and the gyroscope. We assume that the sensor readings are always available. We explicitly consider protecting the privacy of smartphone users' behavioral data. We assume that due to privacy concerns, many smartphone users are not willing to send their sensor data to a centralized authentication service for joint model training with other users' data. We consider both scenarios, where training is done with or without other users' data.

While this paper assumes a single legitimate user of a smartphone, our detection methodology can be extended to allow multiple legitimate users to share a smartphone by deploying multiple models trained for different users.

## 3 ALGORITHMS FOR IMPOSTOR DETECTION

For impostor detection, we explicitly consider three important factors: attack detection capability (security), usability and user data privacy of the solution. We first show that Deep Learning algorithms can work better than the best past work using sensors and Machine Learning, e.g. [12], in the conventional 2-class classification scenario (Section 3.1). We then explore the privacy-preserving scenario (Section 3.2) where only the legitimate user's data is used for training.

### 3.1 Two-class Algorithms and Metrics

A good impostor detection solution needs to be able to detect suspicious impostors while not affecting the usability of the smartphone owner. At the center of this trade-off is selecting an appropriate algorithm for impostor detection. One of our key takeaways is that choosing the right algorithm (and model) is more important for achieving security and performance goals than increasing the model size or adding hardware complexity to accelerate a model.

Previous work on implicit smartphone user authentication mostly leverage (user, non-user) binary classification techniques [6, 12, 31]. This scenario requires both data from the real user and other users for training, and we call it the Impostor Detection-as-a-Service (IDaaS) scenario in this paper. For a specific customer, the data from him/herself are labeled as benign or negative (the user), while all data from other customers are labeled as malicious or positive (non-users). We select certain classification-based machine learning algorithms that give the best accuracy for impostor detection (security) and legitimate user recognition (usability) in the non-privacy-preserving IDaaS scenario. We treat them as benchmarks when comparing with our privacy-preserving impostor detection algorithms.

Among the many Machine Learning (ML) algorithms we investigated, we report the results on Support Vector Machine (SVM) and Kernel Ridge Regression (KRR). SVM is a powerful and commonly used linear model, which can establish a non-linear boundary using the kernel method. KRR alleviates over-fitting by penalizing large parameters and also achieves the highest detection rate in the literature [12] while requiring the computation of 14 heuristically chosen features. Surprisingly, we show that even a simple Deep Learning algorithm (Multi-layer perceptron, MLP) without heuristic and tedious hand-crafting, can outperform it.

**Metrics.** While security is commonly measured as FNR, the percentage of actual attacks that are not detected, we use the inverse term TPR, the percentage of attacks that are detected. Similarly, while usability is commonly measured in FPR, the percentage of normal user attempts that are incorrectly detected as attacks, we use the inverse term TNR, which is the percentage of normal user attempts detected as normal. TPR and TNR enable us to have metrics where higher is always better.

Eq (1) gives the formulae for TPR and TNR, as well as for the other metrics commonly used in comparing the ML/DL models: accuracy, recall, precision and F1. Accuracy is the percentage of correctly identified samples over all samples. Recall, like TPR, is the percent of all attacks that are detected whereas precision is the percent of all reported attacks that are real attacks. F1 is the harmonic mean of recall and precision.

$$
\begin{aligned}
TNR &= \frac{TN}{TN + FP} \\
TPR &= Recall, R = \frac{TP}{TP + FN} \\
Accuracy &= \frac{TN + TP}{TN + FP + TP + FN} \\
Precision, P &= \frac{TP}{TP + FP} \\
F_1\ Score &= \frac{2 \times Recall \times Precision}{Recall + Precision}
\end{aligned}
\tag{1}
$$

### 3.2 Protecting Behavioral Data Privacy

The above binary classification approaches can only be applied to the IDaaS scenarios where the data from other users are available. However, smartphone users may not be willing to send their sensor data to a centralized service for training, due to privacy concerns. Therefore, we need to consider another important scenario where the smartphone user only has his/her data for training. We call this the local anomaly detection (LAD) scenario.

We consider two representative algorithms for one-class learning, i.e. One-Class SVM (OCSVM) and Long Short-Term Memory (LSTM). We propose enhancing the LSTM-based deep learning models with the comparison of reference and actual Prediction Error Distributions (PEDs). We show that *generating and comparing the prediction error distributions is the key to a successful detection for this LAD scenario.* The accuracy of the DL prediction algorithm itself appears to be only of secondary importance.

**OCSVM.** OCSVM is an extension of normal SVM, by separating all the data points from the origin in the feature space and maximizing the distance from this hyperplane to the origin. Intuitively, the OCSVM looks for the minimum support of the normal data, and recognizes points outside this support as anomalies.

**LSTM.** Different from the above discussed stateless models (SVM, KRR, OCSVM, etc.), the LSTM model has two hidden states, $h_t$ and $c_t$, which can remember the previous input information (see

Appendix A for details). We use an LSTM-based model as an outlier detector [17], by training it to predict the next sensor reading, and investigating the prediction errors. The intuition is that an LSTM model trained on only the normal user's data predicts better for his/her behavior than for other users' behavior. The deviations of the actual monitored behavior from the predicted behavior indicate anomalous behavior. Typically, a threshold value is used to decide if the prediction error is normal or not.

**LSTM + Comparing Prediction Error Distributions (PEDs).**
Our intuition is that a single prediction error may vary significantly, but the probability distribution of the errors is more stable. Therefore, comparing the observed PED and a reference PED from the real user's validation data is more stable than comparing the average prediction error with a pre-calculated threshold.

As we do not need to assume the prior distribution of PED, non-parametric tests are powerful tools to determine if two distributions are the same. The Kolmogorov-Smirnov (KS) test is a statistical test that determines whether two i.i.d sets of samples follow the same distribution. The KS statistic for two sets with $n$ and $m$ samples is:

$$D_{n,m} = sup_x |F_n(x) - F_m(x)| \qquad (2)$$

where $F_n$ and $F_m$ are the empirical distribution functions of two sets of samples respectively, i.e. $F_n(t) = \frac{1}{n} \sum_{i=1}^{n} 1_{x_i \leq t}$, and $sup$ is the supremum function. The null hypothesis that the two sets of samples are i.i.d. sampled from the same distribution, is rejected at level $\alpha$ if:

$$D_{n,m} > c(\alpha) \sqrt{\frac{n+m}{nm}} \qquad (3)$$

where $c(\alpha)$ is a pre-calculated value and can be found in the standard KS test lookup table.

### 3.3 Algorithm Experimental Settings

We evaluate the algorithms for impostor detection using the WALK subset in the Human Activities and Postural Transitions (HAPT) dataset [22] at UCI [4]. The HAPT dataset contains smartphone sensor readings. The smartphone is worn on the waist of a group of 30 participants of various ages from 19 to 48. Each reading consists of the 3-axial measurements of both the linear acceleration and angular velocity, so it could be treated as a 6-element vector. The sensors are sampled at 50Hz. We select 25 out of the 30 users in the HAPT dataset as the registered users while the other 5 users act as unregistered users. To investigate the feasibility of user versus impostor classification, the samples from the correct user are labeled negative for impostor detection while all the data from the other 24 registered users and the 5 unregistered users are labeled as positive.

In the IDaaS scenario, each data sample used in both training and testing contains 64 consecutive readings from the same user. At 50Hz sampling frequency, 64 readings correspond to 1.28 seconds which is the latency to detect an impostor. Models are trained for each registered user using his/her data and randomly picked sensor data of the other 24 registered users. We make sure that the training samples have no overlap with the testing samples. The samples from unregistered users are used to examine whether unseen attackers can be successfully detected.

In the LAD scenario, the training data only contains the data from the real user. The testing samples still include the data from the real

**Table 1: Impostor detection in the IDaaS secnario, using binary classification models, achieves 97%-98% accuracy.**

| Models | 64-reading Window | | | | |
|---|---|---|---|---|---|
| | TNR (%) | TPR/Recall (%) | Accuracy (%) | P | F1 |
| KRR | 88.91 | 82.66 | 85.78 | 0.87 | 0.83 |
| ***SVM*** | ***99.26*** | ***97.57*** | ***98.42*** | ***0.99*** | ***0.98*** |
| MLP-50 | 98.31 | 92.70 | 95.51 | 0.98 | 0.94 |
| MLP-100 | 98.60 | 94.65 | 96.63 | 0.98 | 0.96 |
| MLP-200 | 98.41 | 95.72 | 97.06 | 0.98 | 0.97 |
| ***MLP-500*** | ***98.68*** | ***95.47*** | ***97.07*** | ***0.99*** | ***0.96*** |
| MLP-50-25 | 98.13 | 94.49 | 96.31 | 0.98 | 0.96 |
| MLP-100-50 | 98.44 | 95.45 | 96.95 | 0.98 | 0.96 |
| ***MLP-200-100*** | ***98.47*** | ***95.72*** | ***97.10*** | ***0.98*** | ***0.97*** |

user and the other users. We test for window sizes of 64, which is the same size as the IDaaS scenario, and 200, which corresponds to a longer detection latency of 4s but shows how much the detection accuracy can be improved by (Table 2). An LSTM-based model is trained to minimize its average prediction error for each registered user. In the testing of LSTM-based models, prediction errors for consecutive readings in each sample form a testing PED.

### 3.4 Algorithm Evaluation

We evaluate each of the 25 registered users against each of the 30 users, i.e. 750 test pairs in total, and we report the average metrics of all pairs. Table 1 and Table 2 show the results of different algorithms in the IDaaS and the LAD scenarios, respectively. Table 1 shows that in the IDaaS scenario, the SVM model outperforms the other models, including KRR with 14 manually designed features [12], on all evaluated metrics. A simple deep learning model, MLP, performs almost as well, achieving accuracy > 97%. Larger models, e.g. MLP-500 and models with more layers, e.g. MLP-200-100, also slightly lift the accuracy.

Table 2 shows the approaches we evaluated for the LAD scenario, for 2 window sizes of 64 (left) and 200 (right) sensor measurements. For each LSTM algorithm, we also tested different hidden state sizes, from 50 to 500. **LSTM-th** compares the average prediction error in a window with a threshold obtained from the validation set, while **PED-LSTM-Vote** compares the empirically-derived PEDs. We randomly choose 20 samples of prediction errors from the validation set and use them to represent the reference PEDs. In the testing phase, the prediction error distribution of each testing sample is compared to all the reference distributions. The PED-LSTM-Vote models consider a sample as abnormal if more than half of the D statistics of KS-test are larger than the fixed threshold in Eq (3). Table 2 shows the results for the $\alpha$-values that give the best detection accuracy, i.e. $\alpha = 0.10$ for a 64-reading window and $\alpha = 0.05$ for a 200-reading window.

In Table 2, the one-class SVM (OCSVM) achieves average accuracy of 62.9%, thirty percent worse than the 2-class SVM model trained with positive data involved. The LSTM-th models (64-reading window) have an accuracy between 72% and 75%, only slightly better than the one-class SVM model, regardless of the hidden state size. If PED and statistical KS test are leveraged, we see a significant improvement in the detection accuracy up to 87.1% and 90.2% for a 64-reading window and a 200-reading window, respectively.

Table 2: Impostor detection accuracy in the LAD secnario, using one-class models. The numbers next to LSTM-th and PED-LSTM-Vote, i.e. 50 to 500, are the size of hidden states in the LSTM cell. We test some common levels of $\alpha$, i.e. 0.15, 0.10, 0.05, 0.025 [29], and present the best choices for different window sizes in this table.

| 64-reading Window | | | | | | | 200-readingWindow | | | | | | |
|---|---|---|---|---|---|---|---|---|---|---|---|---|---|
| Models | | TNR (%) | TPR/Recall (%) | Accuracy (%) | Avg P | F1 | Models | | TNR (%) | TPR/Recall (%) | Accuracy (%) | Avg P | Avg F1 |
| OCSVM | | 64.24 | 74.19 | 69.22 | 0.59 | 0.65 | OCSVM | | 50.02 | 75.81 | 62.92 | 0.55 | 0.62 |
| LSTM-th | 50 | 79.37 | 65.13 | 72.25 | 0.59 | 0.60 | LSTM-th | 50 | 72.43 | 67.04 | 69.74 | 0.57 | 0.60 |
| | 100 | 78.76 | 66.72 | 72.74 | 0.61 | 0.62 | | 100 | 72.20 | 69.27 | 70.73 | 0.58 | 0.62 |
| | 200 | 78.50 | 69.64 | 74.07 | 0.62 | 0.64 | | 200 | 67.88 | 71.60 | 69.74 | 0.58 | 0.62 |
| | 500 | 79.14 | 70.29 | 74.71 | 0.63 | 0.65 | | 500 | 67.57 | 74.42 | 70.99 | 0.60 | 0.65 |
| PED-LSTM -Vote ($\alpha = 0.10$) | 50 | *85.55* | 83.60 | 84.58 | 0.84 | 0.84 | PED-LSTM -Vote ($\alpha = 0.05$) | 50 | 82.16 | 91.96 | 87.06 | 0.92 | 0.85 |
| | 100 | 87.80 | 85.68 | 86.74 | 0.86 | 0.85 | | 100 | 84.98 | 93.20 | 89.09 | 0.93 | 0.86 |
| | 200 | 89.27 | 85.00 | __87.13__ | 0.85 | 0.87 | | 200 | 88.49 | 92.00 | __90.24__ | 0.92 | 0.89 |
| | 500 | 87.02 | 83.86 | 85.44 | 0.84 | 0.86 | | 500 | 87.14 | 91.16 | 89.15 | 0.91 | 0.88 |

However, the overhead in execution time may increase. In Section 5, we discuss such security-performance trade-offs, which are essential to algorithm selection in practice.

## 3.5 Insights from Algorithm Performance

The results in Section 3.4 show that For the IDaaS scenario, detection in 1.28s with very high sensitivity levels (95%-99%) can be achieved for accuracy, security (TPR) and usability (TNR) when SVM or MLP models are used. In the data-privacy preserving LAD scenario, the detection accuracy, using our LSTM-based models enhanced by collecting error distributions, can reach 87.13% for the same detection latency of 1.28s. If the user allows a detection latency of 4 seconds which is usually not long enough for an impostor to perform malicious operations on the smartphone after stealing it, the accuracy can be increased to 90.24%. Although the accuracy is not perfect, it is comparable to the state-of-the-art one class smartphone authentication using various handcrafted features and complex model fusion [10] in the literature. Also, our privacy-preserving 1-class model achieves better detection accuracy, F1 score, TPN and TNR results than the state-of-the-art 2-class KRR model with hand-crafted features [12] for this data set when both are using a 64-reading window.

**A key contribution we make is to show that it is the Prediction Error Distributions and KS test that provide the significant increase in detection capability.** While tuning the size of deep learning models, e.g. LSTM, has little impact on accuracy, the KS test for PED comparison increases the overall accuracy by +12.4% for the 64-reading window and +19.2% for the 200-reading window. Therefore, we provide the hardware support for generating empirical PEDs and computing the KS statistic in Section 4.4.

## 4 HARDWARE DETECTION MODULE

Our goal is to design a small but versatile hardware module that can be integrated into a smartphone to perform the entire impostor detection, without needing another processor or accelerator. This not only eliminates the network and cloud attack vectors but also reduces the cost to move data and the contention with other applications for computing on the CPU or the GPU. Ideally, the hardware module can read the latest sensor measurements from a buffer so that the main memory does not need to be involved. Our design goals are:

- Suitable for smartphones and other battery and resource-constrained devices,
- Reduced attack surface for better security,
- Flexibility for different ML/DL algorithms and trade-offs of security, usability, privacy, execution time, storage and energy consumption,
- Scalability for more performance in the future if needed.

Unlike prior work on implementing deep learning models in hardware [7, 14, 27], we are interested in neither the highest performance nor the lowest energy consumption. Rather we want to investigate what performance is sufficient with minimum hardware that can achieve an important security goal (like imposter detection), while still being flexible for future needs, such as different algorithms or more performance. To reduce the attack surface, SID should be able to support detection without subsequent processing on another device like the CPU. This includes collecting and comparing empirical probability distributions to enhance DL models. While our primary goal is to perform the best algorithms for impostor detection, namely, MLP and SVM for the IDaaS scenario, and PED-LSTM-Vote for the LAD scenario, we also want SID to be flexible enough to support other ML/DL algorithms as well. For performance scalability, we design SID to allow more parallel data tracks to be implemented, if desired. An innovative aspect of our design is that the SID macro instructions implementing the selected ML/DL algorithm do not even have to be changed when the number of parallel tracks is increased and performance increased. This is in line with our goal of minimal hardware.

## 4.1 Functional Operations Supported

We first consider what operations are needed by the Deep Learning (and Machine Learning) algorithms we want to implement. These are first the PED-LSTM algorithm, and also the MLP and SVM algorithms, which are the best imposter detection algorithms for the two scenarios considered in the previous section. Table 4 shows the operations needed for these different DL/ML algorithms. The instructions from **Vargmax** to **Vsqrt** (at the bottom of Table 4) are needed only by the KRR algorithm [12], the previous highest performing method, to compute the 7 features for the accelerometer and the gyroscope each, i.e., the average, maximum, minimum and variance of the sensor data and three frequency domain features: the main frequency and its amplitude and the second frequency.

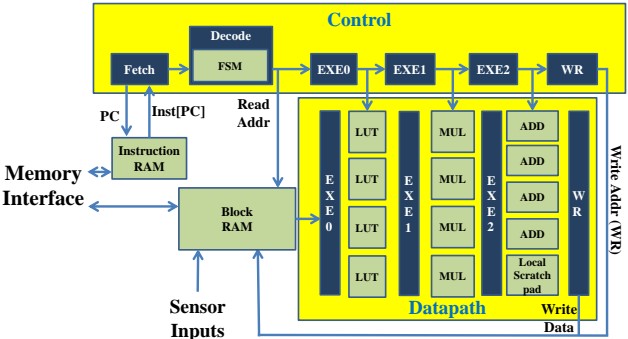

**Figure 1: A SID module implementing vector and matrix operations with 4 parallel datapath tracks.**

| 127 124 | 123 110 | 109 96 | 95 64 | 63 32 | 31 0 |
|---|---|---|---|---|---|
| Mode | Length | Width | Addr_x | Addr_y | Addr_z |
| 4 bits | 14 bits | 14 bits | 32 bits | 32 bits | 32 bits |

**Table 3: SID Macro-instruction with Scalable FSM Control**

We decide not to implement these operations, since they are not needed by the other higher-performing algorithms, while needing significant extra hardware.

## 4.2 Programming Model and Macro-instructions

The programming model of SID is to execute macro-instructions, where each macro instruction is a whole vector or matrix operation. The number of iterations is automatically determined by the hardware, in a Finite State Machine (FSM), based on the hardware's knowledge of the number of parallel data tracks available in the implementation. The macro-instruction supplies the type of operation needed, and the dimensions of the vector or matrix.

The format of a SID macro instruction is shown in Table 3. The **Mode** field specifies one of the operation modes in Table 4. Three memory addresses can be specified in a macro-instruction: **Addr_x** and **Addr_y** for up to two operands, and **Addr_z** for the result. Instead of implementing vector machines with vector registers of fixed length, we use memory to store the vector or matrix operands and results. This design is less expensive than vector registers. It is also more efficient since it supports the flexible-size inputs and outputs and operates seamlessly with our automatic hardware control of the execution of a vector or matrix operation. This is one way (memory not vector registers) to use minimal hardware and achieve scalability (macro-instruction with FSM control).

Each macro-instruction initializes the FSM state of the control unit to indicate the number of iterations of the specified operation. Each cycle, the FSM updates the number of uncomputed iterations, according to the number of parallel tracks, to decide when a macro-instruction finishes (details in the following paragraphs). Thus, the same SID software program can run on SID hardware modules with a different number of parallel tracks, without modification. This achieves our performance scalability goal.

The **Length** and **Width** fields can initialize three state registers, **reg_length**, **reg_width** and **reg_width_copy**, which define the control FSM state. The FSM can be configured by instructions

in two ways: the one-dimension iteration and the matrix-vector iteration. For the one-dimension iteration in a vector operation, the value of **reg_length** is initialized by the **length** field of the instruction. During execution, **reg_length** is decreased every cycle by N(track), which is the number of parallel datapath tracks, until **reg_length** is no larger than N(track) and the next instruction will be fetched.

When an instruction for a matrix-vector operation is fetched, the **length** field initializes **reg_length** and the **width** field initializes both **reg_width** and **reg_width_copy**. A matrix-vector multiplication macro-instruction computes the product of a matrix of size **width**-by-**length** with a vector containing **length** elements. The SID module performs loop tiling by computing a tile of **width** rows by N(track) columns in the matrix before moving to the next tile. When the last tile of columns in the matrix is computed, the next instruction can be fetched.

## 4.3 Parallel Datapaths and Functional Unit Reuse

Figure 1 shows a SID implementation with four parallel datapath tracks. Each track consists of a Look-up Table (**LUT**), a multiplier (**MUL**) and an adder (**ADD**), which are put into three consecutive pipelined execution stages (**EXE0, EXE1 and EXE2**). We also have a small local scratchpad memory in the last execution stage (**EXE2**) for faster access to intermediate results during a macro-instruction. The control path shows 6 pipeline stages: fetch, decode instruction, 3 execution stages and write the result back to memory.

Each macro-instruction is decoded into the FSM control mechanism in the Decode stage of SID's pipeline. This design is scalable since the hardware is aware of the number of parallel datapath tracks that are implemented and can perform automatic control of the FSM for vector and matrix operations, and any loop iterations required. For performance, the control by an FSM avoids using branch instructions for frequent jump-backs in loop-control, as is needed in general-purpose processors, which can take up a large portion of processor throughput for the simple loops needed to implement vector and matrix computation.

We discuss two optimizations: we reduce the number of memory accesses with local storage and we minimize the hardware design cost by reusing functional units.

When computing the matrix-vector multiplication in the **MV-mul** mode, we use a local scratchpad memory and loop tiling to save the latency of storing and accessing partial sums from memory and also reduce the memory traffic. In the **Vmaxabs** mode which finds the maximum absolute value in a vector by doing a comparison of input elements in the EXE2 stage, a temporary maximum is stored in the local scratchpad to reduce external memory accesses; it gets updated every cycle. In the **Vsqnorm** mode which computes the squared L2-norm of a vector, the local scratchpad memory stores the partial sum of $x[i]^2$ (x is the input vector), which are computed by the multipliers and adders in the EXE1 and EXE2 stages.

When computing non-linear functions like sigmoid (**Vsig**), tanh (**Vtanh**) and exponential (**Vexp**), we avoid implementing complex non-linear functional units but use the flexible look-up table (LUT) to look up the slope and intercept for linear approximation. An added benefit of our approach over prior work, e.g. [2], is that

| Operations | Description | IDaaS | | | | LAD | | | | Support Status |
|---|---|---|---|---|---|---|---|---|---|---|
| | | KRR | MLP | SVM Basic | SVM w/ Gaussian Kernel | OCSVM w/ Gaussian Kernel | LSTM | KS-test | LSTM-KS -Mvote | |
| Vadd | Element-wise addition of two vectors | ✓ | ✓ | ✓ | ✓ | ✓ | ✓ | ✓ | ✓ | Yes |
| Vsub | Element-wise subtraction of two vectors | | | | ✓ | ✓ | ✓ | ✓ | ✓ | Yes |
| Vmul | Element-wise multiplication of two vectors | ✓ | | | ✓ | ✓ | ✓ | ✓ | ✓ | Yes |
| Vsgt | Element-wise set-greater-than of two vectors | ✓ | ✓ | ✓ | ✓ | ✓ | ✓ | | ✓ | Yes |
| Vsig | Sigmoid function of a vector | ✓ | ✓ | | | | ✓ | | ✓ | Yes |
| Vtanh | Tanh function of a vector | | ✓ | | | | ✓ | | ✓ | Yes |
| Vexp | Exponential function of a vector | | | | ✓ | ✓ | | | | Yes |
| MVmul | Multiplication of a matrix and a vector | ✓ | ✓ | ✓ | ✓ | ✓ | ✓ | | ✓ | Yes |
| VSsgt | Set-greater-than to compare a scalar and a vector's elements | | | | | | | ✓ | ✓ | Yes |
| Vmaxabs | Find the maximum absolute value of a vector | ✓ | | | | | | ✓ | ✓ | Yes |
| Vsqnorm | Squared L2-norm of a vector | | | | ✓ | ✓ | ✓ | | ✓ | Yes |
| Vargmax | Find the index of the maximum in a vector | ✓ | | | | | | | | No |
| Vmin | Find the minimum in a vector | ✓ | | | | | | | | No |
| Vmax2 | Find the second largest number in a vector | ✓ | | | | | | | | No |
| VFFT | Compute the Fourier transform of a vector | ✓ | | | | | | | | No |
| Vsqrt | Compute the square root of each element in a vector | ✓ | | | | | | | | No |

Table 4: Computation primitives needed by different ML/DL models and statistical testing.

we place the LUTs before the multipliers and adders in the three consecutive execution stages so that no extra multipliers or adders are needed. The ELE0 (LUT) stage of SID outputs a slope, $k[i]$, and an intercept, $b[i]$, for each input value. The interpolation is then computed in the later two stages as $z[i] = k[i] \times x[i] + b[i]$ with $z[i]$ being the value of the non-linear function for input $x[i]$. Also, instead of having another adder tree stage for the **MVmul** mode, we save on hardware cost by reusing the adders in the EXE2 stage to sum the products computed in the EXE1 stage and the partial sum read from the local scratchpad. The new partial sum is written back to the local scratchpad memory if the computation is not finished.

**Integration in smartphone SOC.** We integrate our SID anomaly detection module closer to the sensors to reduce the attack surface and also to save the overhead of memory accesses. (If software processing was used, the sensor measurements would have to be stored to memory first, then read back from memory to the CPU or GPU for software imposter detection.) Modern smartphones have already implemented the interface to write the collected sensor measurements to a cache memory for efficient signal processing [3]. The SID module can leverage a similar interface (viz., "Sensor Inputs" in Figure 1). A valid incoming sensor input can reset the program counter of SID to the beginning of the detection program.

## 4.4 Support for Empirical Distribution Representation and Comparison

Another novel contribution of this work is to show that empirical probability distributions can be collected and compared efficiently using the multipliers and adders already needed for the ML/DL algorithms. To the best of our knowledge, we are the first to describe the following simple and efficient hardware support for collecting error distributions and comparing them with the KS test.

We add two operations for this KS test, but we feel these are general-purpose operations that may also be useful for other ML/DL algorithms and statistical tests as well. The first is a vector-scalar comparison (**VSsgt** described in Table 4). The second operation is **Vmaxabs**, which can be used to find the maximum absolute value in a vector.

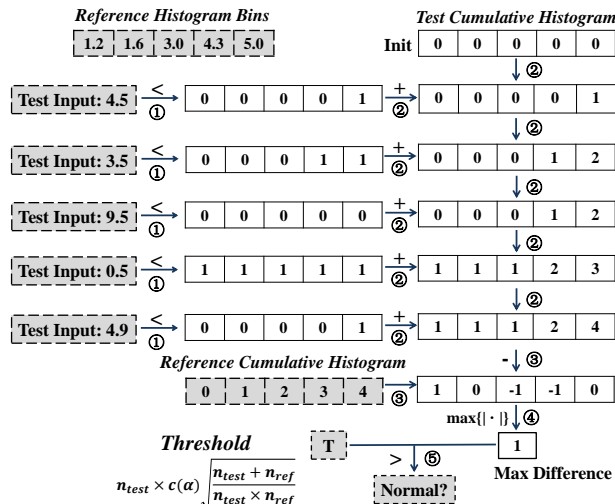

Figure 2: An example of five-step KS test.

We illustrate with an example in Figure 2, showing a five-step workflow. The grey dotted boxes represent inputs, which include the reference prediction error distribution (PED), the test PED, the threshold and the output. The reference PED is collected in the training phase and is represented by reference histogram bin boundaries and a reference cumulative histogram. The test PED is collected online and represented by a series of observed test errors.

Step ①: compare an observed error with reference bin boundaries. The output of this step is a vector of "0"s and "1"s. "1" means that the corresponding bin boundary is greater than the observed error, and "0" otherwise. This uses the **VSsgt** operation. Step ②: accumulate all binary vectors from ①. The accumulated vector, namely the "test cumulative histogram", represents the cumulative histogram of the observed test errors using the reference bins. Step ③ and step ④: find the largest difference in the reference and test cumulative histograms. Step ③ is a vector subtraction and step ④ is the **Vmaxabs** operation, to find the maximum absolute value in

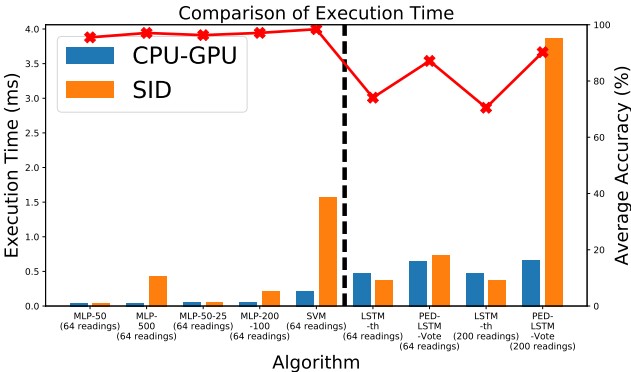

**Figure 3: Execution time and accuracy of detection algorithms. The red line stands for the average accuracy.**

a vector. Step ⑤: compare the largest difference with a threshold, which are both treated as single-element vectors, to determine if the test histogram is normal.

Since the result of Step ④ is the maximum difference in frequency and not normalized, we do the equivalent comparison as Eq (3) by scaling the threshold in Eq (2) up by the number of data points in the testing error distribution ($n_{test}$). In our experiments, $n_{ref}$, which is the number of data points in the reference error distribution, and $n_{test}$ are hyper-parameters that are decided during training and always set to be the same. In the example of Figure 2, both $n_{ref}$ and $n_{test}$ are 5.

## 5 EVALUATION

We evaluate the cost in terms of execution time (performance) and memory usage for the detection algorithms from Section 3 implemented on the SID module. We also show that SID has lower energy consumption and needs much fewer hardware resources than other hardware implementations.

### 5.1 Accuracy vs. Execution Time

New sensor measurements, e.g., for motion sensors like the accelerometer and gyroscope, are available every 20 ms in most smartphones. Hence, we cannot detect any imposter in less than this time. Ideally, we would like the detection mechanism to be performed in less than this time, for the fastest detection.

We consider 3 HW platforms: a CPU-only platform, a CPU-GPU platform and our SID standalone HW module. The CPU-only platform with 32 Intel Xeon E5-2667 cores cannot meet the real-time detection as running a single prediction on one sensor sample with the **LSTM-200** model takes more than 20 ms. The SID module and our **CPU-GPU** platform which has an Nvidia GTX 1080Ti GPU can meet this requirement, so we show only these 2 platforms in Figure 3.

Figure 3 compares different machine learning and deep learning algorithms for their trade-offs between accuracy and execution time on the CPU-GPU platform and SID. The algorithms to the left of the black dashed line are used in the IDaaS scenario. Although the SVM algorithm achieves slightly higher accuracy than MLP-200-100 (98.4% versus 97.1%), it needs significantly longer execution time.

The algorithms used in the LAD scenario are to the right of the dashed line. We measure the execution time of the best algorithm in Section 3.4, i.e. **PED-LSTM-Vote**, and the baseline algorithm, **LSTM-th**, for comparison. We choose the best LSTM size, which is 200, and consider the cases of both the 64-reading window and the 200-reading window. Figure 3 shows that while the KS test technique increases the detection accuracy, it also needs additional execution time. However, the execution time (less than 4 ms) of all algorithms is always much smaller than 20ms, so the performance of SID is more than adequate to achieve imposter detection with the highest accuracy of **PED-LSTM-Vote** at the fastest realtime rate dictated by the sensor measurement speed of 20 ms.

### 5.2 Accuracy vs. Memory Usage

Figure 4 compares models used in the IDaaS and LAD scenarios, in terms of their accuracy and the model size. In the IDaaS scenario (left), we see that a 2-layer MLP-200-100 can achieve a slightly higher detection accuracy with a smaller model size than MLP-500. The SVM model has little improvement on the accuracy over MLP-200-100 but incurs the highest cost in terms of memory usage as it has to store support vectors. Hence, MLP-200-100 appears to be the best for the cost (execution time + memory usage) versus accuracy trade-off for the IDaaS scenario.

In the LAD scenario (right) which is better for protecting the privacy of a user's sensor data, we evaluate the additional memory usage of the KS test technique compared to the baseline **LSTM-th** algorithm where the size of the LSTM cell is still 200. We find that PED-LSTM-Vote uses only 1.6% and 4.8% more space, for the 64-reading window and 200-reading window, respectively. They are better choices in the memory-versus-accuracy trade-off as the improvement in accuracy is significant.

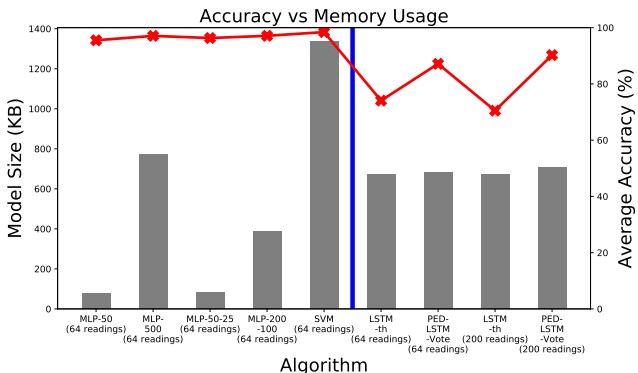

**Figure 4: Model size and accuracy of detection algorithms. The red line stands for the average accuracy.**

### 5.3 Hardware Design Complexity

We implement an FPGA prototype of the SID module, in order to compare with other FPGA implementations. Our implementation has four parallel tracks and a 256-byte scratchpad memory. The size of the datapath RAM is 1.75MB and the size of the instruction RAM is 128KB. We use 32-bit fixed-point numbers, since prior work [14] has shown significant accuracy degradation with 16-bit numbers.

**Table 5: SID hardware requirements and power compared to performance-oriented RNN accelerators**

|  | C-LSTM | DeltaRNN | SID |
|---|---|---|---|
| Functionality | Supports LSTM only | Supports Gated Recurrent Unit only | Supports LSTM, GRU, other ML/DL models and statistical tests |
| FPGA | Xilinx Virtex-7 | Kintex-7 XC7K100 | XC7Z045 FFG900 |
| Quantization | 16-bit fixed point | 16-bit fixed point | 32-bit fixed point |
| Slice LUT | 406,861 (49.07X) | 261,357 (31.52X) | 8,292 (1X) |
| Slice Flip-flop | 402,876 (106.07X) | 119,260 (31.40X) | 3,798 (1X) |
| DSP | 2675 (167.19X) | 768 (48X) | 16 (1X) |
| BRAM | 966 (1.98X) | 457.5 (0.94X) | 489 (1X) |
| Clock Freq | 200MHz | 125MHz | 115MHz |
| Power (W) | Running: 22 | Static: 7.9 Running: 15.2 | Static: 0.12 Running: 0.62 |

The platform board is Xilinx Zynq-7000 SoC ZC706 evaluation kit. The hardware implementation is generated with Vivado 2016.2.

In Table 5, we compare SID implementing LSTM-PED-Vote to two FPGA implementations of Recurrent Neural Network (RNN) accelerators, C-LSTM [27] and DeltaRNN [7]. The C-LSTM algorithm represents some matrices in LSTM as multiple block-circulant matrices, to reduce the storage and computation complexity. DeltaRNN ignores minor changes in the input of the Gated Recurrent Unit (GRU) RNN to reuse the old computation result and thus reduce the computation workload. These two accelerators are capable of RNN inference, but lack the support for generating and comparing empirical prediction error distributions, which we have shown is indispensable to achieve acceptable accuracy.

The FPGA resource usage of Slice LUTs, Slice Flip-flops and DSPs of SID are one or two orders of magnitude less than the other two RNN implementations, which shows a major difference between our minimalist SID module and the performance-oriented accelerators. We measure the FPGA power consumption using the TI Fusion Digital Power Designer tool. The power consumption is an order of magnitude less, making it more suitable for a smartphone.

While we have used an FPGA implementation as a prototype of SID to compare with existing FPGA accelerators, further power reduction is achievable using an ASIC implementation in real smartphone products.

## 6 RELATED WORK

User-behavior-based implicit smartphone authentication has been investigated in the literature. Lee et al. [12] exploited smartphone and smartwatch sensors for user authentication. Frank et al. [6] proposed Touchalytics, a smartphone authentication system based on touchscreen input as a behavioral biometric. Zheng et al. [31] proposed leveraging user tapping behaviors for authentication. These work require other users' data for model training, falling into our IDaaS scenario and raising behavioral data privacy concerns. There have been preliminary works on authenticating smartphone users with only that user's data, which might fit under our LAD scenario. For example, multi-motion sensor [25], fusion of swiping and phone movement patterns [10] and keystroke [9, 13] have been used for one-class smartphone user authentication. However, these works leveraged manually-crafted features without deep learning. In contrast, we show the superiority of deep learning algorithms,

without needing tedious feature extraction, for both the IDaaS and LAD scenarios.

Many accelerators for a single machine learning (ML) algorithm have been proposed, e.g. SVM [19][20], k-neatest neighbors [24] and k-means[1]. Recent work have also been proposed for deep neural networks (DNNs). EIE [8] and Minerva [21] exploit data sparsity of weights and activations during inference to improve performance and energy efficiency. The sparsity in training is exploited in [23] to improve performance. Minerva [21] presents a framework to reduce power consumption by finding the optimal data quantization in the accelerator with software exploration. Weight sharing in CNN is identified by [26] in early software exploration before designing a dedicated accelerator. Some hardware accelerators support multiple machine learning models. [16] evaluates the acceleration of four models in an embedded CPU-GPU-Accelerator system. MAPLE [15] accelerates the vector and matrix operations found in five classification workloads. PuDianNao [14] highlights the non-vector operations and data locality in seven ML techniques. These works support their chosen ML/DL algorithms and exploit different properties of DNN models to improve performance or energy efficiency, while we aim for sufficient performance for real problems at a low cost. They also do not support other analysis techniques such as the collection and comparison of empirical probability distributions with KS tests, as we do.

For minimalist hardware design, we incorporate conventional energy-saving techniques, e.g. the tiling method [2], that can benefit multiple ML/DL algorithms. We do not implement hardware modules for a specific ML/DL algorithm. To the best of our knowledge, we are the first to explore delivering versatility and sufficient hardware performance for detecting anomalous behavior, with reduced energy consumption, rather than shooting for maximum hardware performance or maximum energy efficiency.

## 7 CONCLUSIONS

We study how sensors in a smartphone can be used to detect smartphone impostors and theft using machine learning and deep learning based algorithms. A key contribution is showing that we can detect impostors while preserving the user's privacy by using LSTM-based models enhanced by comparing Prediction Error Distributions (PEDs). In both the IDaaS and LAD scenarios, our deep learning based algorithms have better detection accuracy than the studied machine learning algorithms including the state-of-the-art algorithm with hand-crafted features.

To further reduce the attack surface, we design a low-cost hardware module, SID, to support the best impostor detection algorithms we found in both scenarios. It is versatile enough to support other ML/DL algorithms as well. It has an innovative hardware implementation for collecting and comparing empirical probability distributions that we use to represent user behavior. This enables users of SID to trade-off security with data privacy in choosing one of the two scenarios, as well as choosing trade-offs in accuracy with overhead. Our FGPA implementation shows that SID provides sufficient performance with minimal cost. Compared to other model-specific accelerators, SID provides more functional versatility and uses less hardware resources and energy which are one to two orders of magnitude less than performance-oriented FPGA accelerators.

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

# Appendix A    DEEP LEARNING ALGORITHMS

**Multi-layer Perceptron (MLP)** is a family of feed-forward neural network models, consisting of an input layer, an output layer and one or more hidden layers in between. Formally, the inference of MLP computes:

$$
\begin{aligned}
h_1 &= f(W_1^T x + b_1) \\
&\cdots \\
h_n &= f(W_n^T h_{n-1} + b_n) \\
\hat{y} &= softmax(h_n)
\end{aligned}
\tag{4}
$$

where $h_i$ denotes the output of layer $i$, $f$ denotes a non-linear activation function, e.g. sigmoid or ReLU [1].

**LSTM.** When being used to model temporal sequences, an LSTM cell updates its hidden states $(h_t, c_t)$ for each input time-frame using the previous states $(h_{t-1}, c_{t-1})$ and the current input $x_t$ as described in Eq (5), where the W's and U's are weight matrices, and the b's are bias vectors. Three control gates, the forget gate $f_t$, the input gate $i_t$ and the output gate $o_t$, are used to determine how much of the old states are preserved.

$$
\begin{aligned}
cand_t &= tanh(W_c \times x_t + U_c \times h_{t-1} + b_c) \\
f_t &= \sigma(W_f \times x_t + U_f \times h_{t-1} + b_f) \\
i_t &= \sigma(W_i \times x_t + U_i \times h_{t-1} + b_i) \\
o_t &= \sigma(W_o \times x_t + U_o \times h_{t-1} + b_o) \\
c_t &= f_t \odot c_{t-1} + i_t \odot cand_t \\
h_t &= o_t \odot tanh(c_t)
\end{aligned}
\tag{5}
$$

---

[1] ReLU$(z)$ = $z$ if $z \geq 0$, else 0

