# OpenReview forum: "Smartphone Impostor Detection with Behavioral Data Privacy and Minimalist Hardware Support"
_tinyml.org/tinyML/2021/Research_Symposium — tinyML 2021 Regular_

### Official Review · AnonReviewer4 · 2021-01-26

**Overall Merit Score:** 4

**Brief Summary:**

This is about smartphone imposter in-place detection with single feature from learning from user bhavour. This could be implemented with additional simple independent chip in smart phone as well. Their key contributions are:
- privacy-preserving smartphone imposter detection
- big improvements in accuracy with PED
- Demonstrated simple DNN, KRR with honed feature selection
- Proposed a new lightweight H/W module


**Detailed Comments:**

I'd like to know how a new dedicated H/W co-existing with main SoC in smartphone since usually sensors are routed to main SoC, and how this trend of adding simple secure H/W module would affect on the future design of smartphone SoC.

**Paper Strengths:**

- Clear, well written paper, easy to follow
- The contributions and results are well presented
- Using an open dataset (UCI - HAPT) – allow for reproducing the results
- Threat model definition – allows for good definition of adversary capabilities
- From a privacy perspective, the authors present models that use only local user data and models that use also external (foreign) user data
- Evaluation of algorithms
- Detailed description of hardware implementation (operational, functional and physical)

**Paper Weaknesses:**

These are not exactly weaknesses, but things that can be improved to make the paper even better

- In Section 3.2 it is mentioned that “The accuracy of the DL prediction algorithm itself appears to be only of secondary importance.” – it would be good to expand this statement here, especially that the relation between accuracy and PED is explained much later
- It is mentioned that the attack surface of the HW module needs to be minimized, so the module needs to be placed very close to the sensors. A short mention (maybe already existing solution ) on how to ensure that the sensors’ data are genuine and not replayed (spoofed) is welcomed. We understand that this is outside the scope of the paper, but it will strengthen the “minimize the attack surface” statement.
- It is mentioned that from the 3 HW platforms evaluated, only 2 can meet the 20ms requirement. Can more details be provided on this
- If feasible: a short discussion/summary on the overhead introduced by the SID module in a mobile device (some details are provided already such as power consumption)

**Poster (If Paper Is Rejected):**

1: Yes, ok for poster sesion to nurture work

**Reviewer Confidence:**

4: The reviewer is confident but not absolutely certain that the evaluation is correct

---

### Official Review · AnonReviewer2 · 2021-01-28

**Overall Merit Score:** 3

**Brief Summary:**

This manuscript proposes a deep-learning-based imposter detection system for the mobile device. The authors using the KS test to improve the detection performance under the local and privacy-preserving settings. The authors also designed a dedicated hardware model, namely SID, to accelerate the algorithm with simple pipelines and vector ALUs. The hardware design satisfies the real-time requirement with low complexity.

**Detailed Comments:**

The key advantage of this paper is the joint design of the algorithm and hardware. The authors present a complete solution to the imposter detection problem under the privacy-preserving setting. However, the key issue of this paper is the lack of justification of the necessity of the SID module. The authors do not give detailed information on how the CPU result is measured on the desktop-level CPUs, which makes the comparison in Figure 3 less convincing. The authors should compare the mobile platform like the mobile processor or DSP using an optimized model and inference framework (like TFLite or libTorch) to justify the necessity of introducing a new hardware design.

**Paper Strengths:**

1.	This paper has a strong motivation under the real scenario and provides a complete solution including the co-design of the algorithm and hardware.
2.	The design of SID is tailored for the specific application with low complexity and high efficiency.


**Paper Weaknesses:**

1.	From the algorithm perspective, the novelty is limited. The KS test has been used in anomaly detection of time series.
2.	The execution time comparison is not under fair settings. The authors should also use a mobile processor/accelerator and a compiled model.


**Poster (If Paper Is Rejected):**

1: Yes, ok for poster sesion to nurture work

**Reviewer Confidence:**

4: The reviewer is confident but not absolutely certain that the evaluation is correct

---

### Official Review · AnonReviewer1 · 2021-01-30

**Overall Merit Score:** 2

**Brief Summary:**

The authors propose a mechanism that uses the legitimate user's data to learn about their normal behavior and to proactively detect imposters into users smartphones by integrating a low-cost minimalist hardware module, named SID, embedded into the smartphone to protect user anonymity. Evaluation is performed on HAPT dataset by using LSTM-based models and implemented on FPGAs. Their deep learning algorithms have greater detection accuracy than the studied machine learning algorithms, including state-of-the-art algorithms with handmade features.

**Detailed Comments:**

The objectives of the paper are clear and well thought-out which gives a complete perspective to the reader as to what the authors are trying to accomplish. The authors present several metrics (Precision and F1-score) for software evaluation along with accuracy that provides a holistic overview of the algorithms in place with regards to their performance. The use of KS test and PED is justified and relevant for the application targeted. The increase in accuracy when these are applied further reinforces their utilization.

However, the work fails to provide sufficient support for a number of its presentations.
·         Use of contemporary DL and ML learning models for detection tasks is not new and the only novelty in the work pertains to the use of the statistical tests. Also, the use of statistical procedures like PED in DL applications has already been explored in literature. To this extent, the novelty of the work is limited.
·         The authors propose a low-level framework that mimics the operations of the algorithms in FPGA. Even though it is somewhat an accurate representation, there is no elaboration regarding what happens when dimensionality increases for the inputs. The authors claim that their hardware RTL is scalable but do not justify this claim with proper evidence in the write up or in the figures.
·         Fig. 1 represents a very high-level view of the module; it will be more interesting if the authors presented the underlying logics that is implemented inside the block along with a complete data flow for the FSM.
·         The comparisons in Table 5 are a bit exaggerated because the two architectures that are compared to have different applications for a bigger size of inputs. Hence, the resource utilization comparisons are not exactly on the same level. Also the power consumption comparison is not accurate, where on one side the power of the FPGA chip is compared to the power of the FPGA board from another side.



**Paper Strengths:**

The paper is clear and well written. The introduction, motivation, objective and the method are sound and reasonable.
2. The accuracy of a LSTM or other RNNs to detect irregular users by comparing prediction error distributions.
3. The proposed hardware SID is light-weight, scalable and efficient. With small hardware overhead and being standalone, the user’s privacy is conserved.


**Paper Weaknesses:**

1. Hardware framework requires better visualization and details about the implemented logic. Also it is not overly clear why “the lowest energy consumption” should not be a subject of matter, particularly when the hardware is meant to be embedded on a portable resource-bound device such as a smart phone.
2. The firmware that implements the task is intended to be integrated as a separate hardware on the smartphone for higher security, but what is not clear, is that how such a  firmware, per se, could be updated without being attacked
3.  Hardware comparisons are not justified and the comparisons in table 5 don’t look legit or fair. For example, the power consumption for DeltaRNN is reported for the board+FPGA+fan, and the 7.9W is not the static power, but rather it’s the power of the board when idle and with a fan turned on. It would be better if the authors compared their results on the basis of FPGA to FPGA devices, and not on the basis of one device to another board.



**Poster (If Paper Is Rejected):**

1: Yes, ok for poster sesion to nurture work

**Reviewer Confidence:**

5: The reviewer is absolutely certain that the evaluation is correct and very familiar with the relevant literature

---

### Official Review · AnonReviewer3 · 2021-01-30

**Overall Merit Score:** 4

**Brief Summary:**

This paper demonstrates a method of detecting users of a phone other than the owner using accelerometer and other sensor data. The contributions are:
 -  A method of training purely on a single users data, so that no potentially-personal information is shared beyond the device.
 - Using deep learning rather than traditional ML methods.
 - Applying prediction to the error distributions of the models over time, rather than using the output of a single model.
 - An implementation of a hardware solution on an FPGA.


**Detailed Comments:**

Comments inline above.

**Paper Strengths:**

Strengths discussed by contribution area:

### A method of training purely on a single users data, so that no potentially-personal information is shared beyond the device

This is an important and often-overlooked issue that matters a lot in practical deployments. The ability to train in isolation on a single user's data is an important advantage of this approach over others.

### Using deep learning rather than traditional ML methods

The comparison of DL to traditional ML methods is very helpful, and seems to indicate that 'standard' deep learning models are superior for this domain, which should help others in the field when considering approaches.

### Applying prediction to the error distributions of the models over time, rather than using the output of a single model

The post-processing of model results to produce actionable information (in this case locking a phone against malicious use) is often neglected, so it's refreshing to see so much effort put into this area. The PEM solution seems a genuine advance, based on my knowledge of the field, and I'll be interested to see how this applies to similar anomaly-detection problems.

### An implementation of a hardware solution on an FPGA

Having an end-to-end implementation is a great reassurance that this technique is practical and plausible for real deployments.

**Paper Weaknesses:**

### Evaluation

The HAPT dataset is comparatively small (30 participants) and wasn't originally designed for this task. Evaluations using this are still useful, especially for comparing against other algorithms, but a dataset more customized for the task would be needed to understand how the algorithm performs in a real user scenario. For example, would variations over time of day, or longer periods, cause unacceptable false positives as the user's behavior changes? This is not an issue unique to this paper though, more a comment for future exploration.

### Sensors

It would be interesting to see a discussion of how different kinds of sensors might be useful, like microphones, touchscreen activity, or GPS. Again, not a big flaw in this paper, just an indication of future directions to explore.

**Poster (If Paper Is Rejected):**

1: Yes, ok for poster sesion to nurture work

**Reviewer Confidence:**

4: The reviewer is confident but not absolutely certain that the evaluation is correct

---

### Decision · Program_Chairs · 2021-02-05

**Decision:**

Accept (Regular)

**Comment:**

Congratulations on your paper's acceptance!

Your paper has been accepted as a full-length regular paper.

Please read the reviews carefully and make sure the concerns are addressed in your final submission.

All accepted papers will be given a slot in the TinyML Summit schedule for an oral presentation on Friday, March 26, 2021.

Camera ready instructions will follow soon. All papers will be hosted on arXiv and published papers will have the following header stamp: “Published as a conference paper at TinyML Research Symposium 2021.” The paper will also be presented on the program website.